# Application of Pan-Omics Technologies in Research on Important Economic Traits for Ruminants

**DOI:** 10.3390/ijms25179271

**Published:** 2024-08-27

**Authors:** Zhendong Gao, Ying Lu, Mengfei Li, Yuqing Chong, Jieyun Hong, Jiao Wu, Dongwang Wu, Dongmei Xi, Weidong Deng

**Affiliations:** 1Yunnan Provincial Key Laboratory of Animal Nutrition and Feed, Faculty of Animal Science and Technology, Yunnan Agricultural University, Kunming 650201, China; zander_gao@163.com (Z.G.); yinglu_1998@163.com (Y.L.); mfli_2000@163.com (M.L.); 2022004@ynau.edu.cn (Y.C.); hongjieyun@163.com (J.H.); 15229238680@163.com (J.W.); danwey@163.com (D.W.); 2State Key Laboratory for Conservation and Utilization of Bio-Resource in Yunnan, Kunming 650201, China

**Keywords:** complex traits, omics integration, pan-omics, rumen microbiota, ruminants

## Abstract

The economic significance of ruminants in agriculture underscores the need for advanced research methodologies to enhance their traits. This review aims to elucidate the transformative role of pan-omics technologies in ruminant research, focusing on their application in uncovering the genetic mechanisms underlying complex traits such as growth, reproduction, production performance, and rumen function. Pan-omics analysis not only helps in identifying key genes and their regulatory networks associated with important economic traits but also reveals the impact of environmental factors on trait expression. By integrating genomics, epigenomics, transcriptomics, metabolomics, and microbiomics, pan-omics enables a comprehensive analysis of the interplay between genetics and environmental factors, offering a holistic understanding of trait expression. We explore specific examples of economic traits where these technologies have been pivotal, highlighting key genes and regulatory networks identified through pan-omics approaches. Additionally, we trace the historical evolution of each omics field, detailing their progression from foundational discoveries to high-throughput platforms. This review provides a critical synthesis of recent advancements, offering new insights and practical recommendations for the application of pan-omics in the ruminant industry. The broader implications for modern animal husbandry are discussed, emphasizing the potential for these technologies to drive sustainable improvements in ruminant production systems.

## 1. Introduction

Understanding life phenomena has always been a fundamental pursuit for humanity. The domestication of sheep, goats, and cattle around ten thousand years ago underscored the significant role of ruminants in human survival and development [1,2,3]. Darwin’s theory of natural selection, proposed in “On the Origin of Species”, elucidated the process of species evolution [4]. Mendel’s pioneering experiments on pea plant hybridization revealed the principles of hereditary trait transmission [5], together forming the foundational principles of modern biology. In the 1950s, life sciences rapidly advanced with the discovery of DNA’s double-helix structure, a major breakthrough in biological history that established DNA as the carrier of genetic information [6], and set a global consensus in genetics. In 1977, Frederick Sanger invented the dideoxy sequencing technique (Sanger sequencing), which allowed accurate DNA molecule sequencing and aided in understanding genome structure and function [7]. The same year saw the creation of the first automated sequencer and the sequencing of the first DNA-based genome, the bacteriophage phiX174 [8], comprising 5386 nucleotides and encoding 11 genes. This milestone marked the beginning of modern genomics research.

Entering the 21st century, the advent of second-generation and third-generation sequencing technologies revolutionized genomics studies [9]. These technologies increased sequencing speed and reduced costs while decoding diverse species’ genome maps. The development of genomics [10], transcriptomics [11], proteomics [12], metabolomics [13], and microbiomics [14] provided detailed insights into specific biological processes or systems. However, these single-omics approaches were limited in comprehensively analyzing complex biological regulatory networks.

In recent years, the era of pan-omics has emerged, integrating genomics, transcriptomics, proteomics, epigenomics, metabolomics, and microbiomics to transcend the boundaries between different types of information, thus obtaining more comprehensive biological insight [15,16]. Compared to single-omics research, pan-omics better elucidates the flow of information from genetics and environment to functional outcomes. As significant livestock, the genetic background and metabolic characteristics of ruminants are crucial for production performance [17]. In this context, this review summarizes the development, methods, advantages, and applications of various single-omics and pan-omics technologies in ruminants, aiming to provide references for studying important economic traits in ruminants. 

## 2. Development of Sequencing Technologies for Ruminants

With the continuous advancement and improvement of sequencing technologies, second-generation sequencing technologies (such as 454 sequencing and Illumina sequencing) have emerged, highlighting the limitations of Sanger sequencing. However, Sanger sequencing is still widely used for sequencing individual genes and identifying single nucleotide polymorphisms (SNPs). As the demand for sequencing increased, including higher throughput, longer read lengths, and reduced sample processing, new high-throughput sequencing technologies such as single-molecule sequencing by the PacBio (Pacific Biosciences, Menlo Park, CA, USA) and Nanopore sequencing by ONT (Oxford Nanopore Technologies, Oxford, UK) emerged [18]. Third-generation sequencing technologies allow for a more comprehensive and efficient study of biological processes such as genomics, transcriptomics, and epigenomics, driving significant advancements in livestock research.

Early genome sequencing and assembly of ruminants using second-generation sequencing technologies revealed genetic backgrounds and diversity, providing crucial information for the conservation and management of genetic resources. Notable milestones include the first assembly of the cattle reference genome in 2009 [19], the assembly of yak in 2012 [20], zebu in 2012 [21], the first assembly of the goat genome in 2013 [22], and the assembly of the sheep genome in 2014 [23] (Figure 1). In recent years, research has increasingly focused on hybrid assembly using third-generation and second-generation sequencing technologies to construct high-quality and even telomere-to-telomere (T2T) reference genomes [24,25,26].

## 3. Single-Omics Technologies

The term “omics” is derived from the Greek root “ome” (-ōma), indicating a system’s collective set in Greek and representing a complete whole of a type of substance within a species or individual when used as a suffix [27]. This root has given rise to many derived terms at different levels of research, such as genomics, epigenomics, transcriptomics, proteomics, metabolomics, and microbiomics (Figure 2).

### 3.1. Genomics

Genomics, first introduced by Thomas H. Roderick in 1986, defines the genome as the complete set of genes within an organism [28]. Genes are the basic units of genetic information. Genomics aims to understand the composition, structure, function, and interactions of genomes, as well as their roles in development, growth, function, and evolution. Genomic research can be categorized into de novo sequencing and assembly, whole-genome resequencing (WGS), reduced-representation sequencing, chip, and pan-genomics, depending on the objectives and methodologies. The significant advancements in molecular biology and genetics, particularly after the Human Genome Project successfully mapped the human genome draft sequence [29], have propelled genomics and driven the research in ruminant genomics forward [30].

De novo sequencing provides resources and reference databases for new species or potentially important genes. While it is widely applied in bacteria and small eukaryotes, achieving high accuracy and continuity in sequencing results remains challenging for higher animals like mammals, which have large genomes with abundant repetitive sequences and complex structures [31]. In 2009, the Bovine Genome Sequencing and Analysis Consortium combined bacterial artificial chromosome cloning with whole-genome shotgun sequencing to obtain the first cattle reference genome using a Hereford breed sample [18], which has been recently upgraded to ARS-UCD2.0. The genome of a three-year-old female Yunnan Black goat was assembled, providing the first reference genome for goats [22]. De novo sequencing provides foundational data for gene editing and gene regulation research, although continuous improvements in technology are needed to enhance the resolution of complex genomes.

On the other hand, whole-genome resequencing involves sequencing different individuals of a species against an existing reference genome to analyze genetic variations such as single nucleotide variations (SNVs), copy number variations (CNVs), insertions and deletions (InDels), and structural variations (SVs). The high-quality chromosome-level reference genome of the wild ancestor of domestic sheep, the Asian mouflon, was first de novo assembled. Through extensive resequencing, the structural variation landscape in the *Ovis* and *Capra* genera was comprehensively analyzed, and numerous novel SVs were identified [32]. In a recent study, the genomes of sequenced 30 Chinese indigenous Nanyang cattle and 432 globally representative cattle were analyzed, revealing high genetic diversity and low inbreeding levels in the Nanyang population. Candidate genes related to reproductive performance (Mitogen-activated protein kinase kinase 2, Progesterone receptor, and Genetic suppressor element 1) and immunity (Nuclear receptor coactivator 2, Heat shock factor 1, and Paired box protein 5) were identified through selection signal analysis [33]. Thus, WGS effectively identifies trait-associated genomic regions, providing scientific bases for genetic breeding.

Finally, the concept of pan-genome was first proposed by Tettelin in 2005, referring to the sum of all genes within a species [34]. Pan-genomics involves sequencing and assembling the genomes of multiple strains, integrating and annotating these sequences to obtain comprehensive genetic information, and analyzing genome variations across individuals or populations [35]. Methods for constructing pan-genomics include iterative assembly, de novo assembly of multiple individuals, and graph-based assembly. Iterative assembly was used to construct a bovine pan-genome at the population level, identifying 83 Mb of non-reference sequences beyond the bovine reference genome. The insertion of the Bovine transcriptional activator 1 transposon was discovered to overlap with enhancer and promoter regions in the first intron of the Adaptor protein, phosphotyrosine interaction, PH domain, and the leucine-zipper-containing 2 (*APPL2*) gene, increasing *APPL2* expression and regulating functions related to immune response, taste, cell proliferation, and glucose metabolism [36]. Recently, using graph-based pangenomes combined with HiFi sequencing technology, researchers performed long-read sequencing and haplotype assembly of 24 individuals from seven taurine cattle breeds, revealing that SVs 66 kb upstream of the KIT proto-oncogene caused whitehead depigmentation in cattle [37]. The genomes of 15 different sheep individuals were assembled and annotated using PacBio HiFi sequencing technology, constructing the first sheep pan-genome and identifying 130.3 Mb of novel sequences [25]. These achievements not only deepen the understanding of genetic diversity and evolutionary processes in ruminants but also explore phenotypic and genotypic differences among different breeds and varieties during convergent evolution and domestication.

These studies have propelled the field of ruminant genomics forward. Although de novo assembly of large genomes presents challenges and limitations, ongoing technological advancements are expected to overcome these difficulties, resulting in more accurate and complete genome assemblies. Overall, the genome is the cornerstone of life sciences research, uncovering the mysteries of life and achieving significant breakthroughs in various fields, including livestock production. However, ruminants often possess large genomes and complex repetitive sequences, complicating genome assembly and annotation. Additionally, genomics alone cannot fully explain the functional impacts of genomic variations such as SNPs and CNVs [31]. As technology continues to advance, our understanding of genomes will deepen, promising further innovative applications in the future.

### 3.2. Epigenomics

The term “epigenomics” was first introduced by British developmental biologist Conrad Hal Waddington in 1942 to describe the causal interactions between genes and their products that bring about phenotypes [38]. This field primarily focuses on the genome-wide characterization of reversible modifications of DNA or DNA-associated proteins [39]. Currently, epigenomics investigates the transmission and expression of genetic information, as well as how epigenetic modifications affect biological processes and chromatin structure, utilizing bioinformatics and high-throughput sequencing technologies to study epigenetic modification patterns across the genome [40].

Epigenomic mechanisms mainly include DNA methylation, post-translational covalent modifications of histones, higher-order chromatin structure, and non-coding RNAs [41]. Histone covalent modifications such as acetylation and methylation maintain or suppress transcriptional activity by affecting the chromatin environment surrounding embedded genes [42]. Research has shown that different maternal diets during sheep pregnancy not only influence the expression of imprinted genes in sheep fetuses but also alter the expression of DNA methyltransferases through DNA methylation modifications [43]. Moreover, evaluating the epigenetic characteristics of sperm between high and low-fertility bulls revealed significant differences in methylation levels, with these differential methylation regions potentially affecting key genes related to sperm processing and embryo development [44]. Another study indicated that reduced expression of lncBNIP3 promotes the proliferation of bovine intramuscular preadipocytes, suggesting that manipulating lncBNIP3 expression could improve beef quality and enhance meat trait genetics [45].

Overall, epigenomics regulates organism development, environmental adaptation, and disease occurrence by influencing gene expression and cell differentiation without altering the coding sequence, primarily through specific chromatin remodeling [46]. In ruminants, epigenomic research has significantly contributed to optimizing production performance, improving livestock management, and enhancing disease resistance and adaptability, thereby promoting the sustainable development of the ruminant industry. However, epigenetic markers are dynamic and influenced by various factors such as the environment and developmental stages, and their regulatory mechanisms are complex, making it challenging to determine how specific markers affect gene expression [47]. Future advancements in real-time epigenomic monitoring technologies will enable the tracking of dynamic epigenetic changes, allowing precise dissection of epigenomic regulatory mechanisms.

### 3.3. Transcriptomics

Transcriptomics, also known as gene expression profiling, was first introduced by Velculescu VE [48]. It studies gene expression in cells or tissues using techniques such as gene expression profiling arrays [49], transcriptome sequencing (RNA-Seq) [11], and gene expression sequence tags [50]. RNA-Seq accurately and rapidly measures gene expression and transcriptional activation, allowing for quantitative analysis of transcript presence, identification of new splice sites, and RNA editing sites, as well as qualitative analysis of transcripts and splice sites [51]. Therefore, RNA-Seq is the preferred method for measuring the transcriptome of an organism [52]. The Longissimus dorsi muscle transcriptome profiles of Chinese Merino and Small-tailed Han sheep were first obtained, identifying key regulatory genes Myelin regulatory factor, Glutathione peroxidase 1, and SH3 and cysteine-rich domain 3 involved in muscle growth and development [53]. Another study revealed meat quality trait differences between Yunling and Chinese Simmental cattle through transcriptome analysis of the Longissimus dorsi muscle, indicating higher expression of fat metabolism-related genes such as Aldehyde dehydrogenase 9 family member A1, Acyl-CoA synthetase long-chain family member 5, and Acyl-CoA dehydrogenase medium-chain in Yunling cattle, while glycolysis-related genes such as Phosphoglucomutase 1 and Galactose mutarotase were lower expressed [54]. These studies provide essential biological foundations for understanding meat quality traits and their regulatory mechanisms among different breeds.

Since the introduction of PacBio’s RSII, Sequel, and Sequel II sequencing platforms, transcriptome sequencing has reached new heights, achieving higher sequence accuracy. The advent of full-length transcriptome technology overcomes the limitations of traditional short-read bulk RNA-Seq, which requires fragmentation and bioinformatics assembly, thereby avoiding potential assembly errors and providing more accurate solutions for analyzing alternative splicing and gene fusions [55]. The latest single-cell technologies can more precisely classify different cell subtypes, revealing functional and gene expression differences among different cells [56]. Single-cell sequencing was highlighted by Nature Methods as the most important methodological advancement in 2013, and since the launch of the Human Cell Atlas Project in 2017, single-cell research has rapidly expanded across multiple species, including humans and mice, with comprehensive tissue and organ cell atlases now largely completed [57,58]. Moreover, single-cell technology has extended from model organisms to ruminants. Single-cell RNA sequencing and analysis were conducted on ten metabolic tissues of lactating Holstein cows, constructing a single-cell transcriptome atlas. It was discovered that Interleukin 17F and Interleukin 17A, produced by T helper 17 cells, regulate gene expression in specialized epithelial cells involved in short-chain fatty acids (SCFA) transport in the foregut [59].

While single-cell transcriptome technology provides new perspectives and methods for studying cell transcriptome heterogeneity, its high-cost limits widespread application. Additionally, single-cell transcriptomics mainly focuses on atlas construction, making it challenging to obtain comprehensive gene expression profiles. Combining single-cell with bulk RNA-Seq could provide more complete gene expression data, aiding in future explorations of gene expression and regulatory mechanisms in different species, tissues, or cells of ruminants, thereby offering valuable theoretical bases and information for subsequent breeding efforts.

### 3.4. Proteomics

The concept of proteomics was introduced by Marc Wilkins in 2009 [60]. Since the composition of proteins in an organism changes over time [61], proteomics, the science of studying the composition and dynamic changes in proteins in cells, tissues, or organisms, aims to elucidate protein expression levels, post-translational modifications, and protein-protein interactions [60]. Proteins are the ultimate executors of gene functions and the embodiment of life activities. Through protein-level analysis, cellular activities can be more accurately reflected, effectively elucidating the molecular mechanisms of life processes [62]. Although mRNA expression levels are often used to infer protein expression, this approach overlooks biological processes such as post-translational modifications, protein cleavage, complex formation, and localization, which can result in numerous mRNA transcript variants [63].

Key methods in proteomics include relative and absolute quantification isotope labeling (iTRAQ) [62], stable isotope labeling with amino acids in cell culture (SILAC) [63], and tandem mass tags (TMT) [64]. The iTRAQ method was used to investigate the impact of different grazing intensities on fat deposition in sheep liver, finding that overgrazing affects nitrogen compound metabolism and immune protein levels, shifting energy resources from carbohydrates to proteins, severely impeding sheep nutritional metabolism (protein and lipid) and resulting in growth retardation and hepatic proteomic changes [65]. Plasma proteomics studies have analyzed protein regulation in cows with different reproductive performances during cyclic and pregnant phases, finding that proteins such as breast-cancer-1-associated RING domain 1 and Alpha Fetoprotein were upregulated in cyclic cows, whereas Alpha 2-HS glycoprotein, Clusterin, and Serpin family A member 6 were upregulated in pregnant cows, leading to reduced reproductive performance. Conversely, downregulation of Fibrinogen-like 2 and Zinc finger NFX1-type-containing 1 in pregnant cows enhanced maternal immune response mechanisms, improving embryo implantation success rates [66].

Comprehensive analysis of protein expression levels and functions in different tissues and physiological states of ruminants using proteomics not only identifies key regulatory proteins in physiological processes but also discovers protein markers associated with different traits. This, combined with genetic markers for joint selection and genetic improvement, accelerates the improvement and breeding of ruminant genetic resources. Understanding the changes and mechanisms of trait-regulating proteins is crucial. However, due to the diversity and complexity of post-translational modifications, single proteomics approaches may not fully reflect cellular functional states. Future research will focus more on elucidating protein interaction networks to enhance the understanding of biological processes.

### 3.5. Metabolomics

In 1999, Jeremy Nicholson innovatively proposed the concept of metabolomics, aiming to quantitatively analyze all metabolites in an organism and explore the relationships between metabolites and physiological or pathological states [64]. Metabolites are small molecular compounds produced during metabolism, influenced by both biological and environmental factors [65]. Metabolomics is divided into targeted metabolomics [66], untargeted metabolomics [67], lipidomics [68], glycomics [69], and spatial metabolomics [70]. This field is considered the closest to phenotype, establishing a more direct link between the quantitative and qualitative analysis of metabolites and the phenotype of an organism [71].

Using lipidomics to assess the semen of two groups of bulls with different antifreeze capacities, it was found that most fatty acids in bull sperm are located in the polar lipid portion, such as arachidonic acid and oleic acid, which regulate sperm membrane fluidity and enhance antioxidant properties during freezing and thawing. Additionally, the higher content of branched-chain fatty acids compared to polyunsaturated fatty acids suggests that branched-chain fatty acids may play a role in sperm membrane packaging and fluidity regulation [72]. Similarly, lipidomics analysis of sheep endometrium showed that pregnant sheep had significantly higher levels of ceramides, diglycerides, and non-esterified fatty acids than cycling ewe [73]. A study using metabolomics to quantify serum metabolic changes in ewes from seven days before breeding to seventy days post-breeding identified five biomarkers—L-arginine, urea, methionine, L-lactate, and valine—associated with conception rates and litter sizes [74]. These findings provide important references for understanding the impact of metabolism and lipid composition on ruminant reproduction, aiding in the improvement of reproductive efficiency and health.

However, metabolomics studies are influenced by multiple factors, making it difficult to reflect long-term states solely through metabolomics. The diversity of metabolites and the high technical requirements for quantitative and qualitative analysis pose challenges. Future research may benefit from real-time metabolic monitoring technologies to capture instantaneous changes in metabolite levels, enhancing the understanding of metabolic regulation.

### 3.6. Microbiomics

The concept of the microbiome was first introduced as an ecological term, referring to microbial communities and their associated environments rather than as an “-omics” concept [14]. Today, the broad definition encompasses microbial communities with unique physicochemical properties, functioning as a dynamic micro-ecosystem that interacts with the larger macro-ecosystem, including eukaryotic hosts, across time and scale [75]. With the advent of sequencing technologies, the complex analysis of microbes and their environments can be categorized into methods such as 16S ribosomal RNA gene sequencing [76], metagenomics [77], metatranscriptomics [78]), and metaproteomics [79].

Ruminants possess a complex microbial ecosystem within their rumen, and there exists a symbiotic relationship between the ruminant microbiome and the host animal [80]. Therefore, microbiomics holds unique advantages in ruminant research. Metatranscriptomic studies have found that proteins encoded by the rumen microbiota of Hu sheep, such as Glycoside hydrolase family 5 proteins (Cel5A-h28 and Cel5A-h11), retain enzymatic activity against carboxymethyl cellulose and p-nitrophenyl-β-d-cellobioside, thereby enhancing cellulose degradation [81]. Metagenomic research has shown that high-grain diets reduce the pH of the sheep cecum, alter specific populations of the cecal microbiota, and affect urease pathways that convert pyruvate to acetyl-CoA and urea to ammonia [82]. Metagenomics was utilized to comprehensively analyze the genes and genomes of gastrointestinal microbes involved in the biosynthesis of B vitamins and vitamin K_2_ in ruminants. This revealed the distribution characteristics of vitamin biosynthesis in the gastrointestinal tract and the impact of diet, as well as analyzing the biosynthesis process of vitamin B_12_ by the microbiota [83]. Metaproteomics identified six active ureases in the bovine rumen and found that the flexibility and urease activity of the rumen wall was influenced by β-turns [84].

As microbiomics advances, it not only deepens our understanding of the symbiotic relationship between microbes and host animals and processes such as nutrient digestion and absorption but also provides significant research value in the fields of ruminant nutrition and microbiology. However, while these technologies can determine the composition of microorganisms, single analyses struggle to accurately predict their functions. Current techniques also face challenges in detecting low-abundance components of the microbiome and culturing microbes.

Ruminant life activities involve a complex regulatory process encompassing multiple “-omics” fields, including genomics, epigenomics, transcriptomics, proteomics, metabolomics, and microbiomics. Single-omics approaches may have limitations in explaining certain biological changes, failing to fully reveal the overall biological characteristics of ruminants. Therefore, vertically integrating pan-omics research results to gradually overcome current limitations is essential for a more comprehensive understanding of the regulatory mechanisms and functions in ruminant life activities.

## 4. Pan-Omics Integrated Analysis and Data Integration Methods

Pan-omics, also known as integrative omics, cross-omics, and pan-omics, aims to integrate two or more omics datasets for data analysis, visualization, and annotation. By simultaneously examining data from different layers, such as genomics, transcriptomics, proteomics, and metabolomics, pan-omics approaches provide a more comprehensive perspective on the functioning of biological systems [85,86]. This integrated analysis can uncover associations between different omics data, thus revealing potential biological mechanisms and regulatory networks through the comparison and selection of various omics combinations (Table 1).

Pan-omics research involves multidimensional and complex types of data, and their processing and integration often rely on advanced bioinformatics tools and statistical methods. In recent years, various methods for integrating omics data have emerged, which can be broadly categorized into six types. It is noteworthy that even with the collection of large-scale omics data, different analytical methods may yield varying results (Table 2).

## 5. Application of Pan-Omics Technologies in the Study of Important Economic Traits and Functions of Ruminants

The integration of multi-omics and pan-omics approaches in the livestock sector holds transformative potential for improving both productivity and animal welfare. By offering a holistic view of the genetic, epigenetic, and environmental factors that influence complex traits, these technologies can guide precision breeding programs aimed at enhancing desirable characteristics such as growth rate, feed efficiency, and disease resistance. For instance, the identification of key regulatory networks through pan-omics analysis enables the selection of animals with optimal gene expression profiles for these traits, thus accelerating genetic gains [105]. Moreover, the integration of metabolomics and microbiomics data can inform dietary interventions that optimize rumen function and nutrient utilization, thereby improving overall animal health and reducing the environmental footprint of livestock production [106]. These advancements not only have the potential to increase the sustainability of animal husbandry but also align with growing consumer demand for ethically produced animal products by ensuring better living conditions and health management for livestock [107]. As the livestock industry continues to adopt these cutting-edge technologies, the shift towards a more sustainable and welfare-oriented approach to animal production is becoming increasingly attainable.

### 5.1. Application of Pan-Omics Technologies in the Study of Growth and Reproductive Traits of Ruminants

The growth and reproductive traits of ruminants have long been significant areas of research in animal husbandry. With advancements in science and technology, as well as the application of pan-omics technologies, substantial progress has been made in these fields. Comprehensive research on ruminants using pan-omics technologies enables a deeper understanding of the genetic basis and regulatory mechanisms underlying their growth and reproduction.

Recent studies have utilized whole-genome sequencing data, protein-protein interaction analysis, and functional annotation of candidate genes to identify Lysine methyltransferase 2E, Latent transforming growth factor beta binding protein 1, and Nipped-B-like protein as candidate genes influencing growth traits in Qinchuan cattle, and Thromboxane A2 receptor as a gene related to reproductive functions [108]. Additionally, techniques such as transcriptomics, epigenetics, and proteomics have been employed to explore protein expression changes during growth, gene expression regulatory mechanisms during reproduction, and to reveal key proteins and non-coding RNA elements involved in regulating growth traits, thereby providing crucial insights into the regulatory mechanisms of growth and reproduction. For instance, it was found that heat stress significantly reduces the expression levels of histone modifications (histone H1, H2A, H2B, H4), DNA methylation, and hydroxymethylation in bovine oocytes and embryos [109]. A comparative analysis of genome-wide DNA methylation profiles in fetal and adult Hu sheep muscle revealed significantly lower levels of DNA methyltransferase-related genes in fetal muscle compared to adult sheep. Combining whole-genome bisulfite sequencing and RNA-Seq data identified nine differentially methylated genes, including adiponectin, C1Q, and a collagen-domain-containing gene, Cyclin A2, Integrin subunit alpha 2, and Myogenin, as candidate genes for muscle growth and metabolism in Hu sheep [110].

Furthermore, research on ruminant semen and amniotic fluid components, as well as sperm and egg quality, has uncovered their important roles in reproduction. Studies on feeding management and nutritional regulation are also crucial aspects of growth trait research, aiding in improving growth efficiency and optimizing growth trait performance. RNA-Seq and proteomics were used to reveal pathways involved in vesicle transport, intracellular transport, pathway activation, signal transduction, and energy metabolism in sheep, demonstrating that urine diversion reduces membrane transport rate and amniotic fluid volume, while urine supplementation increases amniotic fluid volume and alters the expression of related regulatory factors. Cytoplasmic dynein light-chain-1 may stimulate urine-derived membrane transport [111]. Additionally, researchers using RNA-seq and microbiome analysis found that feeding corn silage increases antioxidant capacity in cattle but significantly reduces sperm viability and increases sperm abnormality rates. Follicle-stimulating hormone and luteinizing hormone levels in semen are significantly reduced, while inhibin B levels are significantly elevated, negatively affecting semen quality and fertility in bulls [112]. In summary, integrating genomics, epigenomics, transcriptomics, proteomics, and other pan-omics technologies allows for a comprehensive exploration of the genetic basis and regulatory mechanisms of growth and reproductive traits in ruminants. This integrated analytical approach helps uncover finer molecular regulatory networks, providing a theoretical basis for optimizing breeding and growth management practices in livestock.

### 5.2. Application of Pan-Omics Technologies in the Study of Production Performance of Ruminants

Ruminant production performance primarily involves economic traits such as meat, milk, and fur, which provide rich nutritional sources and play vital roles in clothing, footwear, and household products. Research on these traits considers both genetic and environmental factors. In recent years, the application of pan-omics technologies has offered new perspectives on understanding the regulatory mechanisms of ruminant production performance.

In terms of genetic factors, pan-omics studies have identified key genes and regulatory mechanisms affecting ruminant production performance. Joint analysis of transcriptomics and metabolomics has identified functional genes and differential metabolites in buffalo and beef cattle. Nicotinamide nucleotide adenylyltransferase 3 and DTW-domain-containing 1 were among 1800 differentially expressed genes (DEGs) significantly enriched in amino acid metabolism and protein processing pathways, including glycine and serine metabolism. Additionally, 115 differential metabolites were involved in the metabolic pathways of L-aspartic acid, NADP^+^, and glutathione. Comprehensive analysis indicated that genes and metabolites in the niacin and nicotinamide metabolic pathways might influence meat quality differences [113]. In lamb research, proteomic and metabolomic analysis of Callipyge gene mutation (Callipyge) types +/C and non-mutant samples (+/+, C/+, and C/C) revealed the lowest abundance of cytochrome c in +/C samples and high expression of anti-apoptotic proteins Heat shock protein 70, BCL2 associated athanogene 3, and Parkinsonism associated deglycase, resulting in tougher lamb meat [114]. Joint analysis of the transcriptome and DNA methylome of Merino sheep skin tissue identified transcription factors Kruppel-like factor 4, Lymphoid-enhancer-binding factor 1, and Homeobox C13 involved in specific hair follicle morphogenesis and candidate genes such as Sphingosine kinase 1, Growth hormone receptor, and Protein phosphatase 1 regulatory subunit 27 related to wool traits [115]. Similarly, numerous differential genes and proteins were identified in Longissimus dorsi muscle samples of Hu and Dorper sheep through transcriptomics and proteomics, including Alpha-2-HS-glycoprotein, Apolipoprotein H, and Pyruvate dehydrogenase kinase 4 involved in lipid synthesis, as well as Leiomodin 3 and Adenosine monophosphate deaminase 1 regulating muscle growth and development [116]. Research on the regenerative mechanisms of deer antlers, a high-protein, low-fat natural health product, identified 761 antler-specific genes, including the Homeobox D gene cluster, Snail family transcriptional repressor 2, Twist family BHLH transcription factor 1, and SRY-box transcription factor 9, significantly enriched in pathways related to skeletal development, skin development, and neurogenesis. Genes such as Oligodendrocyte transcription factor 1 and Otopetrin 3 were identified as positively selected genes related to antler development, revealing the regenerative mechanism of cervid antlers [117]. These genetic factor-focused studies using pan-omics to analyze economic trait differences contribute to genetic improvement and the enhancement of meat, milk, and wool quality.

Regarding environmental factors, feeding management, and feed composition significantly impact production performance. The integration of feed nutritional value, transcriptomics, metabolomics, and microbiomics was used to study twin sheep under different feeds. It was found that high-fiber, low-protein forage (Ceratoides) significantly increases the essential fatty acid content (linoleic acid and arachidonic acid) in sheep muscle while enhancing rumen microbial diversity [118]. Additionally, Through metabolomics and transcriptomics, it was discovered that the meat quality of castrated Yudong Black goats is affected by 33 DEGs (such as Insulin-like growth factor 1, Cardiac tropoini T, and Protein phosphatase 2 regulatory subunit bgamma) and 279 differential metabolites, impacting fat accumulation, breakdown processes, and overall meat quality [119]. Comparative analysis of metabolites in milk from healthy cows and subclinical ketosis cows using Liquid Chromatography-Mass Spectrometry (LC-MS), Gas Chromatography-Flame Ionization Detector (GC-FID), and Headspace Solid-phase Microextraction/Gas Chromatography-Mass Spectrometry (HS-SPME/GC-MS) revealed that subclinical ketosis cows had significantly higher levels of 11 metabolites, including tyrosine, galactose-1-phosphate, and carnitine, and lower levels of nine metabolites, including creatinine, taurine, and choline, reducing milk nutritional value and increasing odor-related volatile compounds [120].

In conclusion, the application of pan-omics technologies in ruminant production performance research reveals the complex influences of genetic and environmental factors on meat, milk, and fur traits. These studies enhance production efficiency and product quality in animal husbandry, providing scientific support for optimizing breeding and management practices and promoting sustainable development in the livestock industry.

### 5.3. Application of Pan-Omics Technologies in the Study of the Digestive System and Symbiotic Mechanisms of Ruminants

The digestive system of ruminants exhibits unique structural and functional features, with the rumen being a critical component responsible for the fermentation and digestion of food by a complex microbial community [80]. The rumen not only enhances nutritional absorption efficiency and the conversion of indigestible substances but also provides essential immunological support, which is crucial for maintaining the health and production performance of ruminants [121].

Pan-omics technologies have demonstrated immense potential in elucidating the relationship between rumen microbial communities and digestive functions. A combined microbiome and targeted metabolomics analysis of meat quality differences between white and black Tibetan sheep was conducted. The analysis found that shear force and fat content were positively correlated with cholesterol, deoxycholic acid, *Blautia*, and *Erysipelatoclostridium*, and negatively correlated with lactose and *Mogibacterium*. Further analysis showed that total amino acids, N-3 series fatty acids, and cooking loss were positively correlated with D-lactose, taurine, *Odoribacter*, and *Mogibacterium*. Essential amino acids, polyunsaturated fatty acids, and drip loss were positively correlated with cholesterol, deoxycholic acid, L-carnitine, histidine, *Phascolarctobacterium*, *Lachnoclostridium*, and *Akkermansia* [122]. Additionally, rumen microbes significantly influence milk production performance. Untargeted metabolomics and 16S rRNA gene sequencing were utilized to analyze rumen metabolites and microbial composition at different lactation stages in dairy cows. The study revealed that rumen microbial diversity was higher in the early lactation stage than in the peak lactation stage. In early lactation, seven genera, including *Rikenellaceae_RC9_gut_group*, were significantly enriched, while during peak lactation, four genera, including *Succinivrionaceae_UCG-001* and *Candidatus Saccharimonas*, were significantly enriched. Integrated analysis indicated that an increased abundance of *Fibrobacter* could enhance milk phospholipid content by regulating acetate, L-glutamate, and LysoPE (16:0) in the rumen [123]. Research on deer antlers found that acetic acid and propionic acid concentrations in the rumen during the mid and rapid-growth stages of sika deer antlers were significantly higher than in the early growth stage. As the antlers grew, rumen microbial diversity decreased, negatively correlating with SCFA levels. *Prevotella*, *Ruminococcus*, and *Schaedlerella* dominated the rumen microbiota, while untargeted metabolomics analysis showed that γ-aminobutyric acid and creatine concentrations increased gradually during antler growth. Concurrently, amino acid metabolism, including arginine, proline, and alanine, was enhanced [124].

In summary, the regulation of rumen microbial communities not only affects meat quality traits and milk production performance in ruminants but also influences host growth by modulating the rumen microbiota. Although research faces challenges due to the complex structure and function of the rumen, this complexity also endows it with high scientific value. Therefore, an in-depth understanding of the rumen’s structure, function, and mechanisms has become a research focus, providing better guidance and assurance for the health and production performance of ruminants.

### 5.4. Application of Pan-Omics Technologies in Adaptation to Extreme Environments and Diseases Prevention in Ruminants

The adaptability of ruminants to extreme environments and disease prevention are crucial research directions in animal husbandry. In disease prevention, ruminants often face challenges such as gastrointestinal diseases, metabolic disorders, and infectious diseases. Calf diarrhea is frequently associated with intestinal infections, and traditional antibiotic treatments often lead to antibiotic resistance and residue issues. Oral fecal microbiota transplantation was employed to alter the intestinal microbiota of calves (increasing the relative abundance of *Ruminococcaceae*, *Bacteroidaceae*, *Porphyromonadaceae*) and fecal metabolome (reducing fecal amino acid concentrations), effectively alleviating calf diarrhea [125]. This method offers an effective alternative treatment strategy, improving animal health and production performance. An RNA-seq combined with a proteomics study investigated the response of peripheral blood mononuclear cells to infection by different virulence strains of the Peste des Petits Ruminants virus. The results showed that most DEGs were IFN antiviral response genes involved in immune processes such as type I interferon-mediated antiviral responses [126]. Another study, which integrated transcriptomic data from macrophages infected with pathogens (Mycobacterium bovis and Mycobacterium tuberculosis) with GWAS data, identified four key DEGs—STAT1, PIK3IP1, IDO1, and SLAMF1—all involved in immune response, signal transduction regulation, and pathogen control, and all showing significant expression changes across all experimental groups. Additionally, 11 DEGs were found to directly participate in granuloma formation, and 18 DEGs were involved in NF-kB signaling [127]. Integrating pan-omics approaches to reveal key immune response regulators and signaling pathways in ruminant diseases. Combined transcriptomics and genome-wide association studies have explored the genetic basis of bovine mastitis, identifying genes highly associated with susceptibility and resistance to mastitis, such as C-X-C motif chemokine receptor 1, Hematopoietic cell kinase, and Interleukin 1 receptor antagonist [128]. These findings contribute to understanding the genetic mechanisms of mastitis and provide a basis for improving disease resistance.

Fat deposition in the tails of sheep plays a crucial role in adapting to harsh environments, helping them maintain physiological balance under extreme climatic conditions. High-quality genome assembly, structural variation analysis, and graphical pan-genome analysis were used to identify Homeobox B13 (169 bp insertion variant), Platelet-derived growth factor D (*PDGFD*) (867 bp insertion), and the largest insertion/deletion variation of up to 7.7 kb in the intergenic region between Bone morphogenetic protein 2 (*BMP2*) and Hydroxyacid oxidase 1, which are highly correlated with sheep tail traits [25]. Genome-wide selective sweep analysis was combined with transcriptomics and lipidomics of adipose tissue to identify genes involved in tail fat deposition in fat-tailed and thin-tailed sheep. It was found that *BMP2* and *PDGFD* are involved in preadipocyte formation, Vascular endothelial growth factor A regulates terminal adipocyte differentiation, and genes related to lipid synthesis, such as Proteasome 26S subunit, non-ATPase 1 (*PSMD1*), Ectonucleotide pyrophosphatase/phosphodiesterase 2, and Acyl-CoA synthetase long-chain family member 3, play roles in lipid synthesis [129]. These studies reveal the molecular mechanisms of tail fat deposition, providing a deeper understanding of sheep’s survival ability in extreme environments.

In conclusion, integrated pan-omics studies have deeply elucidated the biological mechanisms of disease prevention and adaptation to extreme environments in ruminants. These studies not only enhance our understanding of animal adaptation mechanisms but also provide scientific bases for optimizing livestock production performance and animal welfare.

## 6. Conclusions and Outlook

Despite significant progress in specific fields through the use of individual omics technologies such as genomics, transcriptomics, proteomics, epigenomics, and metabolomics, each technology alone provides limited biological information. This limitation makes it challenging to fully understand the complex biological characteristics of ruminants and the formation mechanisms of their important economic traits. Thus, single-omics studies often have blind spots when interpreting biological phenomena.

The essence of pan-omics integrative analysis is not merely the simple juxtaposition of two or more omics data but the combination of expertise from different disciplines to vertically integrate these data, overcoming the limitations of single-omics approaches. This integrative analysis aims to extract more comprehensive information from limited data by constructing three-dimensional networks of gene expression regulation and linking them to specific biological problems and functions, thereby achieving a deeper understanding of the regulatory and causal relationships among various molecules.

While pan-omics approaches offer unprecedented insights into the molecular mechanisms underlying economically important traits in ruminants, several challenges and limitations must be addressed to fully realize their potential.

(1)Data Integration and Interpretation: The integration of multi-omics data (genomics, transcriptomics, proteomics, metabolomics, etc.) poses significant challenges due to the complexity of biological systems and the sheer volume of data generated. The heterogeneity of data types and the lack of standardized analytical frameworks can lead to difficulties in data harmonization and interpretation, ultimately affecting the reliability and reproducibility of findings [130].(2)Technological and Computational Barriers: Advanced computational tools and resources are required to handle, process, and analyze large-scale omics data. However, the availability of such resources can be limited, particularly in the context of ruminant research, where funding and access to cutting-edge technology may be constrained. Additionally, the development and application of machine learning algorithms for predictive modeling in ruminants are still in their infancy, posing a barrier to the widespread adoption of pan-omics [131].(3)Biological Complexity and Trait Variability: Ruminants exhibit a high degree of biological complexity and trait variability, influenced by environmental factors, management practices, and genetic diversity. This complexity can obscure the relationships between omics layers and phenotypic traits, making it challenging to draw meaningful conclusions. Furthermore, the dynamic nature of the ruminant microbiome adds another layer of complexity to pan-omics studies [132].

However, with continuous advancements in technologies such as artificial intelligence, machine learning, and data analysis, we expect to see more innovative methods to address these challenges. Through these new methods, the biological characteristics of ruminants can be more comprehensively revealed, thereby providing a reliable scientific basis for enhancing production efficiency and animal welfare in the livestock industry.

## Figures and Tables

**Figure 1 ijms-25-09271-f001:**
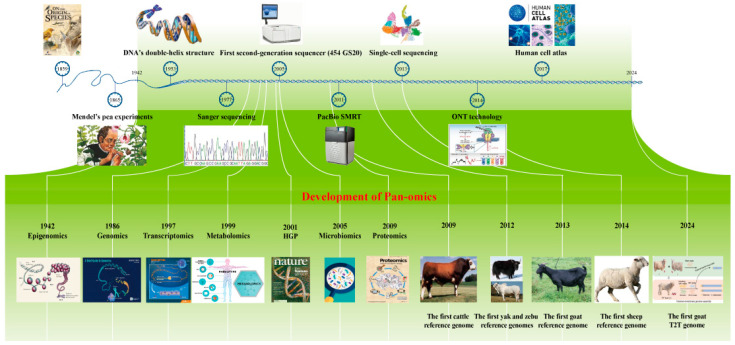
Development of sequencing technology and pan-omics. The timeline in the figure shows the history of genome sequencing technology, key milestone events, and the point in time when the first genome assembly of a ruminant was performed.

**Figure 2 ijms-25-09271-f002:**
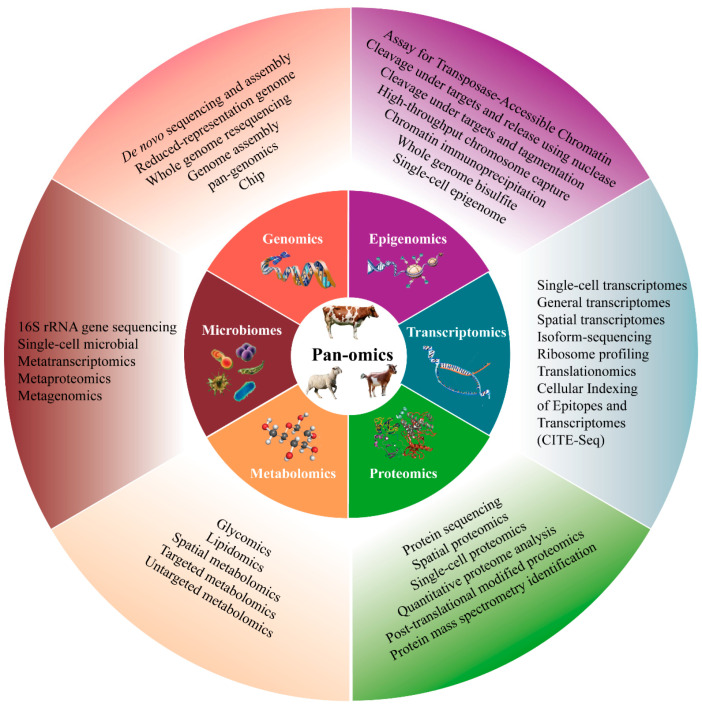
Single-omics techniques and research methods. This radar chart explores relevant research methods in single-omics with a focus on pan-omics.

**Table 1 ijms-25-09271-t001:** Advantages of pan-omics combined analysis.

Omics	Pan-Omics Integrative Analysis	Function	References
Genomics	Transcriptomics	(1) Gene expression regulation: Effects of gene structural variations (SNP, CNV, SV, etc.) on gene expression. (2) Identification of functional variants: Finding gene variants that significantly impact the organism’s function. (3) Developmental and tissue-specific insights: Revealing gene expression patterns in different tissues or developmental stages.	[87,88]
	Proteomics	(1) Correlation between genotype and phenotype and functional validation: Analyzing the relationship between gene variants and protein expression levels and post-translational modifications, and verifying the functional changes at the protein level caused by sequence variations. (2) Complex trait analysis and biomarker discovery: Dissecting the genetic variations in traits caused by changes in protein networks, and identifying protein biomarkers influenced by gene variants.	[89]
	Epigenomics Transcriptomics	(1) Regulation of gene activity and phenotypic variation: Elucidating the effects of DNA methylation, histone modifications, and chromatin accessibility on gene expression and genome stability to obtain different phenotypes. (2) Environmental interactions: Exploring the regulation of gene expression and phenotypic changes by environmental factors.	[90,91]
Transcriptomics	Proteomics	(1) Correlation between mRNA and protein levels and functional validation: Understanding the mechanisms and effects of post-transcriptional regulation of mRNA, and verifying the relationship between gene expression and protein levels and activity. (2) Complex trait analysis: Gaining a more comprehensive understanding of functional protein changes in the regulatory mechanisms of gene expression changes.	[92,93]
	Metagenomics Metabolomics	(1) Regulation of metabolic products by the combined effects of gene expression and the microbiome. (2) Revealing the roles and impacts of microorganisms in host metabolic processes. (3) Discovering the interactions between gene expression processes, the microbiome, and metabolites on the physiological state of the organism. (4) Providing a deeper understanding of the internal regulatory networks and their complexity within the organism.	[93,94]
	Epigenomics	(1) Gene expression regulation: Analyzing how epigenetic modifications (DNA methylation, histone modifications, etc.) regulate gene expression. (2) Understanding cellular differentiation and different physiological mechanisms: Revealing the processes of cell and tissue development and identifying epigenetic dysregulation associated with abnormal gene expression in various physiological states.	[94]
Metabolomics	Proteomics	Analyzing the molecular mechanisms and regulatory pathways of different omics in response to external stimuli, to more intuitively understand the upstream and downstream regulatory relationships between metabolites, enzymes, and genes.	[95]
	Metagenomics	(1) Identifying metabolites produced by the microbiome and their impact on host metabolic pathways. (2) Elucidating the relationship between microbial diversity and function and host metabolic health.	[96,97]

**Table 2 ijms-25-09271-t002:** Advantages and disadvantages of pan-omics data integration methods.

Data Integration Methodology	Advantages	Disadvantages	Typical Methods	References
Connection-based methods	Simple to apply and does not require complex transformations	Increased computational complexity; unable to effectively capture unique structures and relationships of data types	Multi-Omics Factor Analysis (MOFA); iCluster	[98]
Transformation-based methods	Effectively identifies and captures correlations between datasets; simplifies analysis through low-dimensional space	Unable to capture complex biological interactions; the transformation process is computationally demanding	Canonical Correlation Analysis (CCA); Partial Least Squares (PLS)	[99]
Network construction-based methods	Effectively captures interactions and relationships between datasets; visually interprets biological data intuitively	Building and integrating multiple networks is computationally and conceptually complex; limited scalability	Similarity Network Fusion (SNF); Multi-Omics Graph Convolutional Network (MOGCN)	[100]
Matrix factorization-based methods	Effectively reduces dimensions while capturing underlying patterns and structures; can handle various types of omics data and missing values	The decomposed components may be difficult to interpret biologically	Joint Non-Negative Matrix Factorization (NMF); Multi-Omics Tensor Decomposition	[101]
Bayesian and probabilistic methods	Effectively models uncertainty and variability in data; can capture complex, non-linear relationships	Computationally demanding; requires expertise in probabilistic modeling and Bayesian statistics	Bayesian Network Models; Multi-Omics Factor Analysis (MOFA+)	[102]
Deep learning-based methods	Can capture complex, non-linear relationships between omics datasets; automatically learns relevant features and representations from data	High data requirements; deep learning models are challenging to interpret biologically	Deep Learning Autoencoders; Variational Autoencoders (VAE)	[103]
Hybrid methods	Leverages multiple integration strategies to improve overall performance	Increased complexity; balancing and integrating results from different methods is challenging	Integrative Clustering (iClusterPlus); Multi-Omics Factor Analysis via Transfer Learning	[104]

## Data Availability

Not applicable.

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
