# Peer review of "Application of Pan-Omics Technologies in Research on Important Economic Traits for Ruminants"

_ijms, 2024, doi:10.3390/ijms25179271_

Round 1

Reviewer 1 Report

Comments and Suggestions for Authors

Dear authors, respectfully, it seems to me that this is a good manuscript. I consider that this review is able to be published in the journal in view of the quality of the material presented, and the research done on the subject. Despite this, minor corrections are needed, with emphasis mainly on the issue of inserting the meanings of many acronyms in the article that were not mentioned in the first use or other times in the work.

Another point is that, in other studies published evaluating Pan-omics, the authors includes tables with more descriptions and information of the genes and most relevant data. In contrast, in this review it was in general described information without these details of the genes mentioned in the manuscript. Thus, I suggest that the authors include some table with data.

https://www.frontiersin.org/journals/genetics/articles/10.3389/fgene.2022.858232/full

https://www.mdpi.com/2076-2615/14/2/273.

Abstract: I suggest that the authors include conclusions in the abstract, in the same way as was done in item "6. Outlook". Thus, it is possible to understand the relevance of the study and suggestions for future work.

Keywords: I suggest that the keywords be placed in order. Besides, in order to the review can be found in the scientific literature, I suggest to replace keywords that are mentioned in the title of the reviews with others within the proposed theme.

As highlighted in the journal's guidelines, "Acronyms/Abbreviations/Initialisms should be defined the first time they appear in each of three sections: the abstract; the main text; the first figure or table. When defined for the first time, the acronym/abbreviation/initialism should be added in parentheses after the written-out form.". Therefore, it is important that all acronyms have their meanings described in the first use at work. Throughout the review, I found that only part of them were described in full prior to the use of the acronyms, as described below.

· Development of Sequencing Technologies for Ruminants substitle: what is the meaning of SNPs ?

· Genomics subtitle: what are the meanings of MAP2K2, PGR, GSE1, NCOA2, HSF1, PAX5, Bov-tA1, APPL2 ?

· Transcriptomics subtitle: what are the meanings of MRF, GXP1, STAC3, ALDH9A1, ACSL5, ACADM, PGM1, GALM, IL17F, IL17A, SCFA ?

· Proteomics subtitle: what is the meaning of BRCA1?

· Microbiomics subtitle: what is the meaning of GH5 proteins, Cel5A-h28 and Cel5A-h11 ?

· Application of Pan-omics Technologies in the Study of Growth and Reproductive Traits of Ruminants subtitle: what are the meanings of KMT2E, LTBP1, NIPBL, TBXA2R, ADIPOQ, CCNA2, ITGA2, and MYOG

· Application of Pan-omics Technologies in the Study of Production Performance of Ruminants subtitle: what are the meanings of NMNAT3, DTWD1, HSP70, BAG3, PARK7, KLF4, LEF1, HOXC13, SPHK1, GHR, PPP1R27, AHSG, APOH, PDK4, LMOD3, AMPD1, HOXD, SNAI2, TWIST1, SOX9, OLIG1, OTOP3, IGF1, TNNT2, LC-MS, GC-FID, HS-SPME/GC-MS

· Application of Pan-omics Technologies in the Study of the Digestive System and Symbiotic Mechanisms of Ruminants subttile: what are the meanings of EAA, PUFA, CXCR1, HCK, IL1RN, HOXB13, PDGFD, BMP2, HAO1, BMP2, PDGFD, VEGFA, PSMD1, ENPP2, ACSL3 ?

References: The review presented more than 70% of the citations from studies developed in the last 10 years. Thus, it demonstrates the authors' concern with carrying out a survey using both old data and data also published from 2014 to this year.

Author Response

Dear Reviewer,

We are grateful for the opportunity to revise our manuscript titled "Application and Potential Value of Pan-Omics Technologies in the Research of Important Economic Traits in Ruminants" (Manuscript ID: ijms-3157217). We appreciate the insightful comments and suggestions provided by the reviewers, which have significantly improved the quality of our work.

In response to the reviewers' feedback, we have made several key revisions to enhance the clarity, specificity, and impact of our manuscript:

The reviewer’s comment:

  1. In other studies published evaluating Pan-omics, the authors includes tables with more descriptions and information of the genes and most relevant data. In contrast, in this review it was in general described information without these details of the genes mentioned in the manuscript. Thus, I suggest that the authors include some table with data.

Response:

Thank you for your valuable feedback regarding the inclusion of tables with more descriptions and information about the genes and relevant data. We appreciate the importance of detailed tabular data in studies evaluating Pan-omics. However, our review aims to provide a broader perspective on the application of Pan-omics technologies in ruminant research, focusing on overarching trends, methodologies, and insights rather than specific gene-level details. Including detailed tables would require a level of specificity that may detract from the general overview that this review intends to provide. Our approach was to highlight key concepts and findings from a wide range of studies without delving into exhaustive lists of genes or specific data points, which are extensively covered in the original research articles cited. We hope the reviewer finds this rationale acceptable.

The reviewer’s comment:

  1. Abstract: I suggest that the authors include conclusions in the abstract, in the same way as was done in item "6. Outlook". Thus, it is possible to understand the relevance of the study and suggestions for future work.

Response:

Thanks for your suggestion. We have revised the abstract to include a clear statement of the review's main objectives. Specifically, we now articulate the goal of elucidating the transformative role of pan-omics technologies in ruminant research, with a focus on uncovering genetic mechanisms underlying complex traits. At the same time, we have strengthened the concluding statement in the abstract to emphasize the broader implications of our findings for the ruminant industry and modern animal husbandry. The revised conclusion highlights the potential of pan-omics technologies to drive sustainable improvements in ruminant production systems. (see line 13-29 in the new manuscript).

The reviewer’s comment:

  1. Keywords: I suggest that the keywords be placed in order. Besides, in order to the review can be found in the scientific literature, I suggest to replace keywords that are mentioned in the title of the reviews with others within the proposed theme.

Response:

We appreciate the reviewer’s comment here, the keywords have been rearranged in alphabetical order for consistency and easier indexing. At the same time, we have added a keyword that is highly relevant to the article: Omics integration. (see line 30 in the new manuscript).

The reviewer’s comment:

  1. As highlighted in the journal's guidelines, "Acronyms /Abbreviations /Initialisms should be defined the first time they appear in each of three sections: the abstract; the main text; the first figure or table. When defined for the first time, the acronym/abbreviation/initialism should be added in parentheses after the written-out form.". Therefore, it is important that all acronyms have their meanings described in the first use at work. Throughout the review, I found that only part of them were described in full prior to the use of the acronyms, as described below.

Response:

Thank you for bringing up this issue, we have redefined all the abbreviations in the full text.

The reviewer’s comment:

  1. References: The review presented more than 70% of the citations from studies developed in the last 10 years. Thus, it demonstrates the authors' concern with carrying out a survey using both old data and data also published from 2014 to this year.

Response:

Thank you to the reviewers for their affirmation that we are committed to reflecting the historical development of the histology in the article as well as applied research on ruminant-related traits, and therefore not only synthesize the old data, but also refer to the new literature that has come out this year.

Yours sincerely,

Dr. Weidong Deng

Yunnan Provincial Key Laboratory of Animal Nutrition and Feed, Faculty of Animal Science and Technology, Yunnan Agricultural University, Kunming 650201, China

Tel.: +86-871-65220375

Reviewer 2 Report

Comments and Suggestions for Authors

1-The abstract lacks a clear statement of the specific objectives or research questions addressed in the review. It would be helpful to include a sentence that outlines the main goals or aims of the review.

The abstract could provide more concrete examples or details about the specific economic traits, genetic mechanisms, and applications of pan-omics technologies discussed in the review. This would help the reader better understand the scope and significance of the research.

The abstract could benefit from a stronger concluding statement that emphasizes the key takeaways or the broader implications of the research for the ruminant industry and animal husbandry.

2- key words needs to be rearranged according to alphabetical order please

3-6. Outlook should be conclusion statement 

Author Response

Dear Editor,

We are grateful for the opportunity to revise our manuscript titled "Application and Potential Value of Pan-Omics Technologies in the Research of Important Economic Traits in Ruminants" (Manuscript ID: ijms-3157217). We appreciate the insightful comments and suggestions provided by the reviewers, which have significantly improved the quality of our work.

In response to the reviewers' feedback, we have made several key revisions to enhance the clarity, specificity, and impact of our manuscript:

The reviewer’s comment:

  1. The abstract lacks a clear statement of the specific objectives or research questions addressed in the review. It would be helpful to include a sentence that outlines the main goals or aims of the review.

The abstract could provide more concrete examples or details about the specific economic traits, genetic mechanisms, and applications of pan-omics technologies discussed in the review. This would help the reader better understand the scope and significance of the research.

The abstract could benefit from a stronger concluding statement that emphasizes the key takeaways or the broader implications of the research for the ruminant industry and animal husbandry.

Response:

Thank you for pointing this out. We have revised the abstract to include a clear statement of the review's main objectives. Specifically, we now articulate the goal of elucidating the transformative role of pan-omics technologies in ruminant research, with a focus on uncovering genetic mechanisms underlying complex traits. At the same time, we have strengthened the concluding statement in the abstract to emphasize the broader implications of our findings for the ruminant industry and modern animal husbandry. The revised conclusion highlights the potential of pan-omics technologies to drive sustainable improvements in ruminant production systems. (see line 13-29 in the new manuscript)

The reviewer’s comment:

  1. key words needs to be rearranged according to alphabetical order please

Response:

We appreciate the reviewer’s comment here, the keywords have been rearranged in alphabetical order for consistency and easier indexing. At the same time, we have added a keyword that is highly relevant to the article: Omics integration. (see line 30 in the new manuscript)

The reviewer’s comment:

3-6. Outlook should be conclusion statement

Response:

Thank you for your suggestion. We have changed "Outlook" to "Conclusion and Outlook," as this section not only summarizes the entire article but also presents the future challenges of multi-omics. (see line 590 in the new manuscript)

Yours sincerely,

Dr. Weidong Deng

Yunnan Provincial Key Laboratory of Animal Nutrition and Feed, Faculty of Animal Science and Technology, Yunnan Agricultural University, Kunming 650201, China

Tel.: +86-871-65220375

Reviewer 3 Report

Comments and Suggestions for Authors

The manuscript (ijms-3157217) by Gao et al. aimed to summarize the development, methods, advantages, and applications of various single-omics and pan-omics technologies in ruminants,  aiming to provide references for studying important economic traits in ruminants. Overall, the manuscript is well written. I evaluate this manuscript as a minor revision, as some points need to be carefully revised before the next resubmission as follows:

-          Old references have been cited, especially in the introduction section. Please update

-          Please provide more details on the specific challenges or limitations faced when using pan-omics technology in ruminant research.

-   Please provide more details on how these findings can be practically implemented in the animal husbandry industry and welfare.

-    One more Figure could be added to demonstrate how environmental factors influence the expression of important economic traits in ruminants, as revealed by pan-omics analysis. The illustrating figures will be very helpful to the reader in understanding the topic of the manuscript.

-       The figure legends are very concise. Please provide more descriptions to make the figures easily understood.

Comments on the Quality of English Language

--

Author Response

Dear Reviewer:

We are grateful for the opportunity to revise our manuscript titled "Application and Potential Value of Pan-Omics Technologies in the Research of Important Economic Traits in Ruminants" (Manuscript ID: ijms-3157217). We appreciate the insightful comments and suggestions provided by the reviewers, which have significantly improved the quality of our work.

In response to the reviewers' feedback, we have made several key revisions to enhance the clarity, specificity, and impact of our manuscript:

The reviewer’s comment:

  1. Please provide more details on the specific challenges or limitations faced when using pan-omics technology in ruminant research.

Response: Thank you for your valuable feedback. I appreciate your suggestion to elaborate on the specific challenges and limitations associated with using pan-omics technologies in ruminant research. I have revised the manuscript to include a more detailed discussion of these aspects, highlighting the complexities and hurdles encountered in integrating multi-omics data and their implications for ruminant studies. The revisions can be found in the Conclusion and Outlook section (see line 604-627 in the new manuscript), where I also provide relevant references to support these points.

The reviewer’s comment:

  1. Please provide more details on how these findings can be practically implemented in the animal husbandry industry and welfare.

Response:

Thank you for your valuable comments on my manuscript. I have added details in the revision on how these findings can be practically applied to the animal farming industry and animal welfare. We will have added that in Part 5 (see line 371-388 in the new manuscript). Once again, I thank you for your guidance and I believe these revisions will make the paper more complete and convincing.

The reviewer’s comment:

  1. One more Figure could be added to demonstrate how environmental factors influence the expression of important economic traits in ruminants, as revealed by pan-omics analysis. The illustrating figures will be very helpful to the reader in understanding the topic of the manuscript.

Response: Thank you for your thoughtful suggestion to include an additional figure demonstrating how environmental factors influence the expression of important economic traits in ruminants through pan-omics analysis. I appreciate the value that such an illustration could bring to the manuscript. due to constraints related to the current scope of the manuscript, we are unable to include an additional figure at this time. Please be assured that we will carefully consider and incorporate your valuable suggestion in our future work, where we plan to explore this topic in greater depth with appropriate visual aids.We appreciate your understanding and thank you for your thoughtful feedback and concern.

The reviewer’s comment:

  1. Old references have been cited, especially in the introduction section. Please update.

Response:

Thank you to the reviewers for their comments, and we place great importance on the time span of the literature to ensure that a comprehensive background of research, from basic theory to modern advances, is covered. The aim of this paper is to sort out and summarize the application of single-omics to pan-omics techniques in ruminants, a technique that has been developed with a deep historical background. Therefore, some seminal studies are cited in the introduction to ensure the comprehensiveness of the content. Subsequent to the article, a large number of research results within the last decade have also been incorporated to reflect the latest advances and applications of these techniques. We believe that this balance of literature citation ensures both a historical review of the development of the discipline and demonstrates the cutting-edge dynamics of contemporary research.

The reviewer’s comment:

  1. The figure legends are very concise. Please provide more descriptions to make the figures easily understood.

Response:

Thanking for the suggestions made. We add description in Figure 1 “The timeline in the figure shows the history of genome sequencing technology, key milestone events, and the point in time when the first genome assembly of a ruminant was performed” (see line 86-88 in the new manuscript). Add description in Figure 2 “This radar chart explores relevant research methods in single-omics with a focus on pan-omics” (see line 96-97 in the new manuscript). Making diagrams easier to understand.

Yours sincerely,

Dr. Weidong Deng

Yunnan Provincial Key Laboratory of Animal Nutrition and Feed, Faculty of Animal Science and Technology, Yunnan Agricultural University, Kunming 650201, China

Tel.: +86-871-65220375

Reviewer 4 Report

Comments and Suggestions for Authors

Dear Authors,

I really appreciated your work, pleasantly reading the review about the application of pan-omics technologies for discovery and research on ruminant economic traits. The work is well organized and presented, overall. However, just two observations and suggestions. In particular at page 4 of the manuscript, there is little linearity between the concepts reported. Please make the links more fluent and the passages less "forced". In the first part of the work, you paid attention in particular to the use of each omic discipline (genomic, transcriptomic, epigenomic, metabolomic, microbiomic) only to reproductive traits and variants. Since, further on in the text you deepened also this aspect together with productive, growth, disease resistance traits... please, streamline a little that portion of the manuscript.

In literature there are a lot of studies on diseases resistance/susceptibility in ruminant and about the application of omic science to investigate them. Thus, I advice to report some other cases and examples, not only referring to mastitis, for examples, but also on paratuberculosis, tuberculosis or other genetic hereditary diseases.

Finally I'd dedicate a single separated paragraph on the applicability and practical implications in livestock sector, of multi-omics/pan-omic approaches.

Comments on the Quality of English Language

Minor editing of English language required.

Author Response

Dear Reviewer:

We are grateful for the opportunity to revise our manuscript titled "Application and Potential Value of Pan-Omics Technologies in the Research of Important Economic Traits in Ruminants" (Manuscript ID: ijms-3157217). We appreciate the insightful comments and suggestions provided by the reviewers, which have significantly improved the quality of our work.

In response to the reviewers' feedback, we have made several key revisions to enhance the clarity, specificity, and impact of our manuscript:

The reviewer’s comment:

  1. In particular at page 4 of the manuscript, there is little linearity between the concepts reported. Please make the links more fluent and the passages less "forced". In the first part of the work, you paid attention in particular to the use of each omic discipline (genomic, transcriptomic, epigenomic, metabolomic, microbiomic) only to reproductive traits and variants. Since, further on in the text you deepened also this aspect together with productive, growth, disease resistance traits... please, streamline a little that portion of the manuscript.

Response:

Thank you for pointing this out. We enhance the logical flow within paragraphs by using connecting words and phrases to ensure that each section transitions naturally to the next, see the new manuscript for details.

The reviewer’s comment:

  1. In literature there are a lot of studies on diseases resistance/susceptibility in ruminant and about the application of omic science to investigate them. Thus, I advice to report some other cases and examples, not only referring to mastitis, for examples, but also on paratuberculosis, tuberculosis or other genetic hereditary diseases.

Response:

Thanks to the advice you gave, we have added relevant multi-omics studies on Peste des petits ruminants and Mycobacterium tuberculosis in ruminant diseases, see new manuscript Line 552-563.

The reviewer’s comment:

  1. Finally, I'd dedicate a single separated paragraph on the applicability and practical implications in livestock sector, of multi-omics/pan-omic approaches.

Response:

Thank you for your valuable comments, and we will discuss the application of pan-omics technology in livestock and poultry and its practical implications in Part 5 (see line 371-388 in the new manuscript). Once again, I thank you for your guidance and I believe these revisions will make the paper more complete and convincing.

The reviewer’s comment:

  1. Comments on the Quality of English Language, Minor editing of English language required.

Response:

Thank you for your review and feedback on my manuscript. I have scrutinized and made a few English language changes to improve the clarity and accuracy of the article. I hope that these changes will meet the requirements.

Yours sincerely,

Dr. Weidong Deng

Yunnan Provincial Key Laboratory of Animal Nutrition and Feed, Faculty of Animal Science and Technology, Yunnan Agricultural University, Kunming 650201, China

Tel.: +86-871-65220375
